# Reinforcement of Acrylamide Hydrogels with Cellulose Nanocrystals Using Gamma Radiation for Antibiotic Drug Delivery

**DOI:** 10.3390/gels9080602

**Published:** 2023-07-26

**Authors:** Alejandra Ortega, Silvia Valencia, Ernesto Rivera, Tania Segura, Guillermina Burillo

**Affiliations:** 1Departamento de Química de Radiaciones y Radioquímica, Instituto de Ciencias Nucleares, Universidad Nacional Autónoma de México (UNAM), Ciudad Universitaria, Ciudad de México 04510, Mexico; alejandra.ortega@nucleares.unam.mx (A.O.); 17e50265iqe@centla.tec.nm.mx (S.V.); 2Instituto de Investigaciones en Materiales, Universidad Nacional Autónoma de México (UNAM), Circuito de la Investigación Científica, Ciudad Universitaria, Ciudad de México 04510, Mexico; riverage@unam.mx; 3Departamento de Madera Celulosa y Papel, Centro Universitario de Ciencias Exactas e Ingenierías, Universidad de Guadalajara, Las Agujas, Zapopan 45200, Mexico; eikkaas@hotmail.com

**Keywords:** cellulose nanocrystals, AAm hydrogels, gamma radiation, crosslinking, nanocomposites

## Abstract

In this paper, we report the synthesis of acrylamide hydrogels (*net*-AAm) reinforced with cellulose nanocrystals (CNCs) using gamma radiation, a powerful tool to obtain crosslinked polymers without the use of chemical initiators and crosslinking agents. Some slight changes in the chemical structure and crystallinity of CNCs took place during gamma irradiation without affecting the nanofiller function. In fact, cellulose nanocrystals had a notable influence over the swelling and mechanical properties on the reinforced hydrogels (*net*-AAm/CNC), obtaining more rigid material since the Young compression modulus increased from 11 kPa for unreinforced *net*-AAm to 30 kPa for *net*-AAm/CNC (4% *w*/*w*). Moreover, the studies of retention and release of ciprofloxacin (Cx), a quinolone antibiotic drug, showed that reinforced hydrogels were able to load large amounts of ciprofloxacin (1.2–2.8 mg g^−1^) but they distributed 100% of the drug very quickly (<100 min). Despite this, they exhibited better mechanical properties than the control sample, allowing their handling, and could be used as wound dressings of first response because they can absorb the exudate and at the same time deliver an antibiotic drug directly over the injury.

## 1. Introduction

Hydrogels are three-dimensional structures of hydrophilic polymers with a porous structure that have been used in drug delivery applications due to their ability to retain large amounts of aqueous fluids, similar consistency to human tissues, and porous structure [1]. One important property for the successful function of hydrogel in the biomedical field is biocompatibility, because noncompatible materials can generate inflammatory responses in vivo and thus limit their use in living systems [2]. This property may be modified by the selection of the starting materials of hydrogel. There are many hydrogels that exhibit good compatibility using natural polymers as gelatin or chitosan, but the semi-synthetic and synthetic polymers have also shown promising results [3]. Poly(acrylamide) is derived from acrylamide monomer, which in recent years has been used as a biomaterial. In fact, the acrylamide hydrogel (*net*-AAm) is a material widely used in biomedical applications as a carrier for the delivery of drugs and bioactive compounds [4], implants [5], and scaffolds [6], among others since it presents good biocompatibility in vitro and in vivo systems [2,7].

However, the main disadvantage of hydrogels is their weak mechanical strength, which has limited their use [8]. To overcome this problem, a large variety of strategies have been used to improve their mechanical properties, highlighting the use of interpenetrating networks (IPNs) and fillers such as carbon nanotubes (CNTs), metallic nanoparticles, and nanofibrils, among others [9,10].

Cellulose nanocrystals (CNCs) are a filler composed of the crystalline nanostructures of the ordered regions of cellulose, a natural biopolymer made up of glucose repeating units with 1,4-β-glycosidic linkages [11]. These nanostructures have a road-like shape consisting of a cellulose chain crystalline structure and have received extensive attention in the field of polymer science since they possess high mechanical properties, giving them potential applications as reinforcers [12,13,14]. A CNC can vary in its crystallinity, morphology, and physical and chemical properties depending mainly on the cellulose source, which can be natural sources (palms or cotton), agricultural residues (cane bagasse or agave bagasse), or commercial microcrystalline cellulose (CMC) [15,16]. The most employed procedure for obtaining this is the hydrolysis of cellulose with sulfuric acid to remove amorphous regions and release the crystallites. However, not only are the crystallites extracted, but they are also functionalized with sulfur negatively charged ester groups, which facilitate the formation of stable colloidal aqueous suspensions [15,17]. Therefore, CNCs have many advantages over other reinforcement nanofillers because they are inexpensive, abundant, biodegradable, and non-cytotoxic natural polymers [12]. The most important aspect is that they contain sulfur ester groups on their surface, allowing a good dispersion within the hydrophilic polymer matrix since the mechanical and thermal properties are strongly related to the obtained morphologies [18].

Currently, there are many methods for synthesizing the hydrogel, but one of the most used is the chemical process (free radicals, redox, and condensation reactions), which involves the use of initiators and crosslinking agents [19,20]. The final product must be carefully purified to avoid chemical residues that could be toxic and cause biocompatibility problems when hydrogels want to be used for biological, medical, or food applications [21]. Among the chemical methods, hydrogels can also be synthesized through high energy (electron beams or gamma rays), which has the advantage that the crosslinking reaction proceeds without chemical reagents at any temperature, obtaining a pure product as well as the formation and sterilization of hydrogel in a single step [22,23]. Moreover, in comparison to the electron beam, gamma radiation is unidirectional and has a greater penetration power, which allows it to obtain hydrogels with a more homogenous pore size, which has an impact on the swelling and mechanical properties of the material [24,25].

Here, we present the synthesis of acrylamide hydrogels (*net*-AAm) that are reinforced with cellulose nanocrystals isolated from commercial CMC (Sigma Aldrich) with the aim of combining the benefits of gamma radiation with the qualities of a natural biomaterial. Therefore, the gamma radiation effects on the CNC in the obtention of *net*-AAm were studied by the direct method, varying the irradiation dose (10, 15, 20, and 25 kGy) and CNC concentration (0, 1, 2, and 4% *w*/*w*). In addition, the reinforced hydrogels were characterized and tested as drug delivery systems using ciprofloxacin, a broad-spectrum antibiotic employed to treat tract, urinary, skin, and bone infections.

## 2. Results and Discussion

### 2.1. Effect of Radiation in the CNC

To know the gamma radiation effect on the crystallinity and chemical structure of cellulose nanocrystals, an aqueous dispersion of CNC (4% *w*/*w*) was irradiated at 25 kGy (the maximum dose used in this work) and a dose rate of 9.38 kGy h^−1^. The diffractograms of the CNC before and after irradiation are compared in Figure 1, where it was possible to recognize the diffraction peaks of the 2θ angle at 15.5°, 22.5°, and 34°, corresponding to the (110), (200), and (040) crystalline planes that are characteristic of cellulose I [26]. The intensities of these peaks were higher and narrower before irradiation, meaning that the gamma radiation decreased the crystallinity of CNCs from 75% to 67% (after irradiation). This phenomenon occurs because the CNC suffers degradation under gamma radiation by the scission of glycosidic bonds and the aperture of glucopyranose rings, forming carbonyl groups (aldehyde and carboxylic acid) in polymeric chains as well as gaseous products [27,28].

These chemical changes could be appreciated in the IR spectra (Figure 2). CNC (0 kGy) exhibits its characteristic peaks at 3331, 2899, 1647, 1058, and 1030 cm^−1^, associated with the vibrations of O–H stretching, C–H stretching; the O–H bending of absorbed water; and the C–O and C–C stretching of pyranose rings, respectively [29,30]. The irradiated CNC (25 kGy) showed the same absorption peaks, but an additional small band was appreciated at 1745 cm^−1^ due to the formation of carbonyl groups (C=O) in the polymer structure caused by the degradation, which was confirmed by the presence of another peak at 820 cm^−1^ (overtone). This result agrees with previous reports that showed the appearance of carbonyl groups when the CNC was exposed to ionizing radiation and that the concentration was proportional to the applied dose [28,31].

### 2.2. Hydrogels Reinforced with CNC

The AAm-reinforced hydrogels were obtained by adding CNC and irradiating at different doses using the direct method. AAm was 100% reticulated at 15 kGy for all concentrations tested. Data are not shown, but it is important to note that the polymerization and crosslinking of AAm were independent of the added CNC, despite the possibility that a small part of the cellulose could be grafted or copolymerized with the AAm polymer chains since free radicals are generated in the structures of CNC and AAm at the same time during irradiation [32]. This means that the presence of nanocrystals did not interfere with the crosslinking mechanism since all reinforced hydrogels showed a 100% gel yield, like the control hydrogel *(net*-AAm) (Figure 1a). A possible explanation for this is that the cellulose chains are stabilized by inter- and intramolecular hydrogen bonds, but when the material is irradiated at doses higher than 10 kGy, these interactions begin to break down and cause the degradation of cellulose since the free radicals are produced in its structure, generating the scission of glycosidic units (Figure 1b) [33,34]. In this work, the degradation of cellulose was minimal, as shown in the XRD and FTIR of irradiated CNC.

### 2.3. FTIR Determination

The incorporation of CNC into the AAm network was confirmed by FTIR-ATR (Figure 3). CNC showed its characteristic peaks at 3302 cm^−1^, corresponding to stretching vibrations of hydroxyl groups (O–H) at 1644 cm^−1^ due to O–H bending of bound water, at 1428 and 1315 cm^−1^ associates of CH_2_ asymmetric bending and wagging of pyranose ring, respectively; at 1163 cm^−1^ due to C–O antisymmetric stretching; at 1104 cm^−1^ corresponds to C–O stretching; and two sharp peaks at 1058 and 1029 cm^−1^ for stretching vibrations of C–O–C and C-C of glucoside ring, respectively [35,36].

For *net*-AAm, it was possible to observe the vibrations of N–H stretching at 3319 and 3182 cm^−1^ (asymmetrical and symmetrical), C=O stretching at 1647 cm^−1^ (amide I), N–H bending at 1602 cm^−1^ (amide II), and C–N stretching at 1410 cm^−1^ [37,38]. The main absorption peaks of CNC and *net*-AAm were similar, so they overlapped in spectra of reinforced hydrogels (*net*-AAm/CNC), but the signals related to stretching and deformation vibrations of the pyranose ring could be observed. Thereby, *net*-AAm/CNC showed new peaks at 1428, 1316, 1161, 1106, 1083, and 1056 cm^−1^, corresponding to C–H, C–O, C–O–C, and C–C of the glucose ring of the CNC [36,39]. These results confirm the incorporation of the cellulose nanocrystals into AAm hydrogels.

### 2.4. Thermal Behavior

DSC and TGA analyses were carried out for all synthesized hydrogels, but only the results of the samples with 0 and 4% *w*/*w* of CNC are shown (Figure 4). The CNC displayed two endothermic transitions, the first corresponding to the evaporation of moisture (60–140 °C) and the second at 240 °C, which was associated with the melting point [40]. The control hydrogel exhibited a glass temperature transition (T_g_) at 131 °C and another transition at 301 °C. The thermogram of the reinforced hydrogel (4% *w*/*w*) proved the incorporation of CNC in the AAm network structure since it showed a T_g_ at 159 °C. This shift at higher temperatures is attributed to the decrease in the free volume in the network since the OH groups of the nanofiller interact with the NH_2_ groups of AAm through hydrogen bonds, thus restraining the mobility of the polymeric chains [41,42]. Moreover, the reinforced hydrogel showed an extra peak at 247 °C, which is associated with the melting point of the CNC [43].

The thermal decomposition of samples is shown in Figure 5. CNC exhibited a mass loss by the evaporation of bound water that corresponded to 4.8%. Then, it showed two degradation steps at 267 and 367 °C due to glycosidic bond cleavage catalyzed by acid sulphate groups in its structure [36]. The remains were 47.5%, indicating a high amount of carbon in its structure. For *net*-AAm, it was also possible to observe two decomposition steps at 306 and 397 °C due to the elimination of ammonia and the depolymerization of polymeric chains [36,43]. *net*-AAm/CNC (4% *w*/*w*) displayed similar degradation behavior to the AAm network because the CNC concentration was low, but three degradation stages could be distinguished at 306, 371, and 397 °C, the first and third corresponding to the *net*-AAm, and the second to the CNC. The residue of the reinforced material was 23%. The incorporation of CNC into AAm hydrogels increased the residue. It is worth mentioning that thermograms of the reinforced hydrogels with lower CNC percentages are not shown since all of them exhibit the same behavior.

### 2.5. Scanning Electron Microscopy

The cross-section morphology of hydrogels was elucidated by SEM, and the corresponding images are shown in Figure 6. It was possible to observe that the unreinforced hydrogel (Figure 6a) displayed thin walls and a porous structure, but the pores seemed to collapsed, because in a swollen state, the material is very fragile. However, the reinforced samples exhibited a well-defined dense structure with thicker walls and a more homogenous pore distribution (Figure 6b–d), indicating that the CNC helped to obtain a more ordered network, thereby preventing the collapse of pores and enhancing the mechanical properties. These observations have been reported for other reinforced hydrogels with CNCs [44,45]. Moreover, the pore size was affected by the amount of added CNC, showing a decrease in its size as the CNC concentration increases. This could be due to the hydrogen bond interactions between the NH and OH groups of AAm and cellulose, as the DSC results showed.

In addition, micrographs confirmed that the CNCs were not grafted on the AAm polymeric chains since the reinforced hydrogels showed as smooth a porous structure as the control sample (0% CNC), while grafted networks usually look rougher with thin fibers joining the polymeric chains [38,46]. Therefore, the CNC was fully integrated into the AAm network when gamma radiation was used to synthesize the hydrogels, thus confirming that it did not interfere with the polymerization and crosslinking processes of AAm.

### 2.6. Swelling Behavior

The main characteristic of hydrogels is their ability to retain a large amount of water or aqueous solutions [46,47], so it is important to characterize this property. Figure 7 shows the swelling of hydrogels at different doses and CNC concentrations. For *net*-AAm and *net*-AAm/CNC (1% *w*/*w*) irradiated at 10 kGy, it was impossible to determine the swelling ability since they did not have a suitable consistency for handling. The hydrogels obtained with 15 kGy could already be manipulated, and the swelling ability was measured. For *net*-AAm/CNC (2% *w*/*w*) and *net*-AAm/CNC (4% *w*/*w*), the swelling was determined for all used doses (even at 10 kGy) because the CNC concentration was higher, probing its reinforcement effect. Moreover, it was possible to observe that swelling was higher for hydrogels at lower irradiation doses. It is well known that AAm is polymerized because gamma radiation interacts with the double bonds (C=C) through the generation of free radicals. Once the gelation dose is reached, the radiation energy produces the reticulation by abstracting hydrogen atoms from AAm polymeric chains, and the crosslinked density increases proportionally to the irradiation dose [48,49]. It is important because the synthesized hydrogels at low doses swelled more due to the fact that the crosslinking density was lower (Table 1). However, the samples synthesized at 20 and 25 kGy, regardless of the amount of CNCs added, did not present significant changes in the swelling behavior. Therefore, the best dose to obtain the reinforced AAm hydrogels is 15 kGy because the resulting gels showed high swelling and good handling.

On the other hand, the presence of the CNC also influenced the swelling behavior since it interacted through hydrogen bonds with the AAm, causing a restriction in the polymeric chain’s mobility. Therefore, the reinforced hydrogels showed less swelling that *net*-AAm for any irradiation dose (Figure 8). However, at concentrations higher than 2%, a few changes were detected in the swelling behavior because the amount of CNC prevents the interaction of AAm with water molecules. Thus, the maximum swelling percentage of the reinforced hydrogel was dependent on the used dose and the CNC.

The time to reach the maximum swelling was also affected by the CNC concentration since the intermolecular forces between the OH and NH groups hindered the extension of the chains. It was possible to observe that during the first 10 h, the swelling rate increased and then leveled off for the reinforced hydrogels, but the swelling of the control sample (*net*-AAm) continued until 24 h.

### 2.7. Mechanic Properties

CNC was used with the purpose of improving the mechanical properties of AAm hydrogels, and the results of the compressive stress–strain curves are presented in Figure 9. The reinforced hydrogels were able to sustain a higher strain because the Young compression modulus increased from 11 kPa for *net*-AAm to 30 kPa for *net*-AAm/CNC (4% *w*/*w*), which means a change of 173% (Figure 9 and Table 2). These results showed that the CNC concentration substantially improved the strength of the AAm hydrogels. All reinforced hydrogels withstood a higher compressive stress than the control sample, incrementing from 15.7 kPa for *net*-AAm to 433.4 kPa for *net*-AAm/CNC (4% *w*/*w*), which allowed their handling without breaking. The improvement in mechanical properties was associated with the CNC since these get better at higher concentrations because of the strong adhesion between the functional groups of AAm polymeric chains and the incorporated CNC [50,51].

It is worth noting that although slight changes in the CNC crystallinity took place during gamma irradiation (Figure 1 and Figure 2), the influence over the mechanical properties of AAm hydrogel was notorious.

### 2.8. Load and Release Studies

The nanocomposite hydrogels were tested as drug delivery systems using ciprofloxacin (Cx) as the model drug, showing good results in the loading of this broad-spectrum antibiotic (Figure 10). Control hydrogel (*net*-AAm) and the composite with 1% *w*/*w* of CNC showed a similar load profile, retaining between 1 and 1.2 mg of CX per gram of hydrogel. In cases where the composite had a higher CNC concentration (4% *w*/*w*), this value increased up to 2.8 mg per gram, most likely because the main interaction of Cx with *net*-AAm is through hydrogen bonds between the amine, carboxylic acid, fluorine, and amide groups. This interaction increases when the CNC is added due to the large amount of hydroxyl groups that it has, which improves the loading of the drug (Figure 2). Despite these results, the release was similar for all hydrogels regardless of CNC content since they distributed 100% of the drug in the first 100 min of the study. This may be because the drug was mostly retained on the material surface and its availability was immediate. However, the Cx loading was higher than other polymeric materials reported in the literature, where the retention was on the order of micrograms and still exhibited good antimicrobial activity against *S. aureus* and *E. coli* [52,53]. Thus, the synthesized material has potential application as the primary material response in an emergency, such as dressing wounds, where the main complications are related to nosocomial infections [54].

## 3. Conclusions

Reinforced hydrogels of acrylamide with cellulose nanocrystals (*net*-AAm/CNC) were successfully synthesized by gamma radiation, demonstrating that the use of γ rays is an easy method to carry out the polymerization and crosslinking of AAm at the same time since the mechanism of reaction was not affected when low concentrations (1, 2, and 4% *w*/*w*) of nanofiller were used. The IR, DSC, and TGA studies confirmed the incorporation of CNC into the AAm hydrogels, while the SEM micrographs showed that the reinforced hydrogels exhibited a denser structure with thicker walls due to hydrogen bonding interactions between the functional groups of CNC and the AAm polymer chains. The swelling studies showed an inversely proportional relationship between the maximum swelling and CNC concentrations. However, the CNC had a notable influence over the mechanical properties, providing more rigid hydrogels when the CNC amount was higher, which caused the Young compression modulus to increase from 11 kPa for unreinforced *net*-AAm to 30 kPa for *net*-AAm/CNC (4% *w*/*w*). This mechanical improvement allowed hydrogels to be manipulated without breaking. Studies of the retention and release of ciprofloxacin showed that the reinforced hydrogels were able to load large amounts of this drug (1.2–2.8 mg g^−1^) since the main interaction with the material was with hydrogen bonds and the nanofiller contains a large amount of hydroxyl groups. However, the release of Cx was very fast (<100 min) in all the tested samples because the retention was mainly on the surface. Despite this, the reinforced hydrogels have a potential application as the primary material response in an emergency, such as wound dressing as a first response, because they can absorb the exudate and at the same time deliver an antibiotic drug directly over the injury.

In future work, it is necessary to carry out assays of cytotoxicity to elucidate the biocompatibility of reinforced hydrogels and studies of biodegradability to know the influence of CNC on this property.

## 4. Materials and Methods

Cellulose microcrystalline (Avicel pH_101, 50 μm particle size), ciprofloxacin, and acrylamide (99%) were purchased from Aldrich-Sigma México and used without previous purification. Sulfuric acid (98%) was acquired from Meyer Company. Dialysis tubing cellulose membranes with a molecular weight cut-off (14,000 Da) were used to purify CNC, and distilled water was used in all experiments.

### 4.1. CNC Synthesis

The CNC was obtained from CMC from previous reports with slight modifications [55,56]. Briefly, CMC (1 g) was mixed with a sulfuric acid aqueous solution (25 mL, 45% *v*/*v*) and kept with vigorous stirring at 800 rpm for 75 min at 60 °C. After the hydrolysis reaction, the mixture was added over ice water (75 mL) and it was refrigerated for 12 h, allowing it to settle overnight. Then, it was decantated and centrifuged at 20,000 rpm for 15 min. The supernatant was decanted, and the precipitate was suspended in distillated water. The suspension was dialyzed with cellulose membrane dialysis against distilled water for several days until a neutral pH was reached, changing water every 24 h. The colloidal suspension was centrifuged, frozen with liquid nitrogen, and lyophilized.

### 4.2. Preparation of Reinforced Hydrogels

Aqueous solutions of acrylamide (4.8% *w*/*w*) with different concentrations of CNC (0, 1, 2, and 4% *w*/*w*) were prepared. The mixtures were dispersed using an ultrasonicate (700 Hz Vibra cell Sonics) for 20 min at 20% amplitude, with an ice bath to avoid the heating of monomer solution.

A total of 4 g of each mixture was poured into glass ampoules, sealed with septa capes, and purged with argon for 15 min. Samples were irradiated at different doses (10, 15, 20, and 25 kGy) using gamma radiation of a Co^60^ source (Gamma Beam 651 PT from Nordion Co., Ottawa, ON, Canada) with a dose rate of 9.38 kGy h^−1^. After irradiation, the hydrogels were washed with distilled water for one week, changing the solvent every 24 h to eliminate homopolymer without crosslinking. Finally, the samples were filtered and dried at 30 °C in a vacuum oven until constant weight was achieved. The crosslinking percentage was calculated by means of Equation (1).
(1)$$Gel\;(\%)=\frac{W_{f}-W_{i}}{W_{i}}\times 100$$
where W_i_ and W_f_ are the initial and final weight of samples, respectively.

### 4.3. Characterization

#### 4.3.1. X-ray Diffraction

Diffraction patterns were recorded on X-ray diffractometer Bruker D8 Advance (Bruker, Karlsruhe, DE) equipped with a CuK sealed tube X-ray source (λ = 1.54 nm) operating at 30 kV–30 mA. The samples were analyzed under diffraction angle 2θ scanned in the range from 10 to 60° with a step size of 0.02° and a counting time of 3 s. The crystallinity index (Ic) of the evaluated samples was calculated by Segal method [22] using Equation (2).
(2)$$Ic\;(\%)=\frac{I_{200}-I_{am}}{I_{200}}$$
where I_200_ denotes the maximum intensity of the (200) lattice peak at about 22° and correspond to the crystalline part, and Iam represent the amorphous part of cellulose (baseline) at about 18°.

#### 4.3.2. Infrared Spectroscopy

The Attenuated Total Reflectance Fourier Transform Infrared (ATR-FTIR) spectra of the samples were recorded with a Perkin-Elmer Spectrum 100 spectrometer (Perkin Elmer Cetus Instruments, Norwalk, CA, USA), with 16 scans taken over the range from 4000 to 650 cm^−1^.

#### 4.3.3. Thermal Properties

Thermogravimetric analysis (TGA) was carried out using a TGA Q50 (TA Instrument, New Castle, DE, USA) with a heating rate of 10 °C min^−1^ under inert atmosphere, from ambient temperature to 800 °C, whereas for differential scanning calorimetry (DSC), a TA Instrument 2010, at a heating rate of 10 °C min^−1^ under dynamic nitrogen atmosphere. The studied temperature range was from 25 °C to 250 °C.

#### 4.3.4. Scanning Electron Microscopy

Microscopy images were obtained by Field Emission Scanning Electron Microscope JSM-7600F (JEOL, Cambridge, UK) with high resolution at an accelerating voltage of 10 kV. Before the analysis, small pieces of hydrogels were swollen, cryo-fractured from liquid nitrogen, and lyophilized. Then, the cross-section of samples was coated with carbon by sputtering.

#### 4.3.5. Swelling Studies

A small piece of dried sample was weighed and immersed in distilled water at room temperature. The swollen hydrogel was taken out, superficially cleaned with dry paper to remove excess water on the surface, and weighed. The procedure was repeated at different time intervals until constant weight was observed. Finally, the percentage of swelling was calculated using Equation (3).
(3)$$\mathrm{Swelling}\;(\%)=\left(\frac{W_{t}-W_{0}}{W_{0}}\right)\times 100$$
where W_t_ is the weight of the sample in the swollen state at any specific time, and W_0_ is the initial weight of the dry sample.

#### 4.3.6. Mechanical Tests

The compression assays were carried out on the hydrogels with different amounts of CNC in the maximum swelling state using a universal testing machine from Shimadzu AGS-X/100N (Shimadzu, Kyoto, Japan). Cylindrical samples (10 mm × 10 mm) were compressed with a speed of 3 mm min^−1^ and the maximum deformation reached was 80%. The temperature during experiment was maintained at 25 °C. The compressive elastic modulus (E) was calculated with strain measurements from 10 to 15%, while stress (ϭ) and strain (ɛ) from measured force and displacement were based on the initial radius and height of samples. The assays were carried out in triplicate.

#### 4.3.7. Load and Release Studies

The loading and release studies were carried out using ciprofloxacin (Cx) as drug model. Little pieces of samples (10–20 mg) were immersed in 10 mL of an aqueous solution of the drug (0.0135 mg mL^−1^) and placed under constant stirring at 25 °C. Small aliquots were taken at different time intervals and the change in absorbance was measured by UV spectroscopy at 265 nm (UV-Vis spectrometer SPECORD200 Plus, Analytik Jena, Thuringia, Germany). The experiment was carried out in triplicate with independent samples, and the results are shown in retained mg of Cx per gram of hydrogel (mg/g).

The drug-loaded samples were removed from solutions, gently rinsed with water, and the excess was dried with a paper towel. Then, they were dried in a vacuum oven for 24 h at room temperature.

For the release study, the loaded and dried samples were placed in 10 mL of buffer saline solution (PBS, pH: 7.4) at 37 °C with magnetic stirring to simulate physiological conditions. The absorbance change was also monitored at different time intervals by UV-Vis spectroscopy; the experiment was carried out in triplicate, and the results are shown in retained mg of Cx per gram of hydrogel (mg/g).

## Data Availability

The data of this research will be available on request from correspond author.

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
