# Peer review of "Reinforcement of Acrylamide Hydrogels with Cellulose Nanocrystals Using Gamma Radiation for Antibiotic Drug Delivery"

_gels, 2023, doi:10.3390/gels9080602_

Round 1
Reviewer 1 Report
Overall, the manuscript carries an excellent idea. The article has been well thought and written. The data is mainly good and many readers might be interested in these results. The paper is interesting and worth publishing.Author Response
The authors appreciate the time dedicated to review this manuscript and the favorable comments. Thank you very much!!
Reviewer 2 Report
Alejandra Ortega et al. reported “Reinforcement of Acrylamide Hydrogels with Cellulose Nano-crystals using Gamma Radiation for Antibiotic Drug Delivery”. The present work is publishable in Gels after addressing the following major issues.
1. In the abstract part no need of mentioning characterization technique. Revise the abstract and include the main finding of the present work. characterization technique should be mention in the conclusion part. Avoid to write abbreviation in the abstract section.
2. Explain the safety measurement of gamma rays in the introduction section.
3. I don’t think so that Acrylamide is biocompatible and the present study is about biomedical application. Explain it with references.
4. Explain other synthesis methods for the synthesis of hydrogel like redox, UV, etc. and why gamma is preferred over others method. Read and cite the following literature which will helpful to improve the introduction section. https://doi.org/10.3390/gels9010064.
5. What is the mechanism of gamma radiation polymerization (acrylamide with cellulose in the presence of H2O)? Include in the revised manuscript.
6. If possible, include the XRD of Reinforcement of Acrylamide Hydrogels with Cellulose Nano-crystals in the revised manuscript for comparison.
7. Draw the mechanism of interaction of drug molecules with polymeric network functional groups in the revised manuscript.
8. Check the loading capacity at different pH range in the revised manuscript.
9. Conclusion need improvement.
10. Check the grammatical and spelling mistakes throughout the manuscript.
Moderate editing of English language required
Author Response
The authors appreciate the time dedicated to review this manuscript and the comments. Please see the attachment

Reviewer 3 Report
The paper submitted by Ortega et al. deals with the synthesis of composite hydrogels reinforced with cellulose nanocrystals. Adequate tools have been used for the formulation and the physico-chemical characterization to obtain crosslinked composite hydrogels with “green” processes.
Effects of gamma irradiation on the composite hydrogels have also been explored.
Relating to the application, the studies of retention and release of ciprofloxacin, antibiotic drug, showed preliminary results for wound dressing.
The paper is well presented and structured and could be published in Gels according the following minor revisions.
1) check English grammar and vocabulary in the whole MS, especially in the introduction sections.
2) Thermal Behaviour:I did not see the control CNC 4% alone: it would be useful to be added.
3) Swelling behavior : Control 0 kGy should be useful.
4) P10 line 266:please precise again the expected results.
5) Load and release studies: please eleasing profile in mg/g and precise the % of CX released? Does "mg/mg" refer to mg CX released/g total CX initially added? It should be better elucidated in the method section
6) Conclusion: please discuss biodegradability of the hydrogel system with or without cellulose?
English should be improved
Author Response

(The authors gave the same response as above.)

Round 2
Reviewer 2 Report
The author addressed all the issues in the revised manuscript and now its publishable in the present form.
Although, for safety measurement of gamma-rays read the following manuscript.
Journal of Hazardous Materials Advances 10 (2023) 100299 (https://doi.org/10.1016/j.hazadv.2023.100299)
OK